# Hoarseness, Quality of Life, and Social Anxiety: A Case–Control Study

**DOI:** 10.3390/bs15091160

**Published:** 2025-08-26

**Authors:** Süleyman Dönmezdil, Serdar Ferit Toprak

**Affiliations:** 1Department of Psychology, Mardin Artuklu University, Mardin 47070, Turkey; 2Department of Audiology, Faculty of Health Sciences, Mardin Artuklu University, Mardin 47070, Turkey; sftoprak@hotmail.com

**Keywords:** hoarseness, dysphonia, quality of life, social anxiety, voice disorders, anxiety

## Abstract

Hoarseness is a common voice symptom that can impair communication and lead to psychosocial difficulties. It has been hypothesized that chronic hoarseness may contribute to elevated social anxiety. This study aimed to assess the impact of hoarseness on quality of life and social anxiety in affected individuals. Thirty-eight patients with chronic hoarseness (voice disorders) and 40 matched healthy controls were evaluated in a prospective case-control study. Quality of life was measured using the WHOQOL-BREF questionnaire (Physical, Psychological, Social, and Environmental domains). Social anxiety was assessed with the Liebowitz Social Anxiety Scale (LSAS), and general anxiety and depression with the Hospital Anxiety and Depression Scale (HADS). Group scores were compared using appropriate statistical tests, and effect sizes with 95% confidence intervals were calculated. Patients with hoarseness had significantly lower Psychological Health and Social Relationships scores on the WHOQOL-BREF than controls (*p* < 0.01 for both; large effect sizes), indicating worse quality of life in these domains. Physical Health and Environmental domain scores did not differ between groups. The hoarseness group also showed higher social anxiety: LSAS total scores and Social Interaction subscale scores were significantly greater than those of controls (*p* < 0.01 and *p* < 0.05, respectively; moderate-to-large effects), whereas the Performance Anxiety subscale was similar between groups. By contrast, HADS anxiety and depression scores did not differ significantly between patients and controls. Notably, mean HADS scores in both groups fell in the mild (borderline) range rather than the normal range. Chronic hoarseness is associated with reduced quality of life in emotional and social domains and with increased social anxiety symptoms, but not with elevated general anxiety or depression. These findings underscore the need to address psychosocial factors, particularly social anxiety, in the clinical management of patients with voice disorders.

## 1. Introduction

Hoarseness is a common symptom of dysphonia (voice disorders), characterized by noticeable deviations in voice quality, pitch, or loudness relative to a person’s age, gender, or cultural expectations ([1]). Voice disorders are quite common in the general population; the lifetime prevalence of voice disorders in adults currently approaches 30% and the point prevalence at any given time is around 6–7% ([16]; [5]). Certain occupations that heavily rely on voice use—such as teachers, call-center operators, and singers—carry an even higher risk of developing voice problems ([16]; [20]; [17]; [9]; [24]), and may experience greater voice-related psychological distress ([13]). Individuals with dysphonia often struggle to communicate as a result of their impaired voice, which negatively impacts their social lives, job performance, and overall quality of life ([12]). Similarly, a primary care study found that dysphonia had a significant self-reported impact on daily life ([6]). Patients with voice disorders experience disruptions in their social life and report symptoms of psychological distress (such as stress and anxiety) that are directly related to their vocal impairment ([12]). The challenges posed by chronic hoarseness are not limited to communication difficulties, but can also have significant psychosocial consequences. Current research indicates that patients with voice disorders often experience psychosocial distress—such as depression and anxiety—in addition to physical health problems ([25]; [10]). Importantly, the link between emotional distress and voice disorders can be bidirectional: emotional factors (e.g., stress or personality traits) may contribute to the development or exacerbation of dysphonia, just as voice problems can lead to psychological distress. The deterioration of one’s voice can lead to insecurity about speaking in social settings and evoke fears of embarrassment or negative evaluation. In the long term, this may predispose individuals to developing social anxiety disorder (social phobia). Social anxiety disorder is characterized by marked fear or anxiety in social situations where one might be scrutinized by others, manifesting as significant discomfort during activities like conversing, public speaking, or meeting new people ([11]; [22]). Individuals with hoarseness may develop avoidance behaviors in social settings due to worries that their voice will not be understood or will be perceived as “strange.” Therefore, examining the relationship between hoarseness and levels of social anxiety is clinically important.

Prior studies have shown that individuals with voice disorders tend to exhibit higher levels of depression and general anxiety compared to those without voice problems ([2]). However, specific aspects such as social anxiety have not been well studied in this population. To address this gap, we conducted a case–control investigation focusing on psychosocial outcomes in dysphonic patients. In the present study, we aimed to evaluate the extent and specific domains in which quality of life is affected in individuals with chronic hoarseness and determine whether their social anxiety levels are elevated, as well as whether their general anxiety and depression symptoms differ, compared to matched healthy controls. We hypothesized that, relative to the controls, patients with hoarseness would have a poorer quality of life in psychosocial domains (particularly the psychological and social relationship domains of the WHOQOL-BREF) and higher social anxiety levels, but would not show significantly higher general anxiety or depression. This hypothesis was based on the expectation that voice-related challenges predominantly affect emotional and social functioning (e.g., communication confidence) rather than causing broad psychopathology. By including outcome measures spanning multiple domains—a general quality of life scale, a specialized social anxiety scale, and a general anxiety/depression scale—our study was designed to clarify whether the psychosocial impact of hoarseness is context-specific or part of a more generalized psychological effect. The findings of this study could shed light on the importance of psychosocial support in the treatment of hoarseness and help increase healthcare professionals’ awareness of this issue.

## 2. Materials and Methods

### 2.1. Study Design

This research was designed as a prospective, case–control observational study. Approval was obtained from the appropriate institutional ethics committee (Diyarbakır Gazi Yaşargil Training and Research Hospital Ethics Committee, date of approval 24 July 2020, No. 534) and written informed consent was obtained from all participants in accordance with the Declaration of Helsinki.

### 2.2. Participants

The patient group consisted of 40 adult patients who presented in otorhinolaryngology clinics with chronic hoarseness (defined as voice complaints lasting at least 3 months). Two patients were excluded due to incomplete questionnaire data, yielding a final patient sample of 38 individuals (20 women, 18 men; mean age 45.3 ± 12.1 years). The control group included 40 individuals (22 women, 18 men; mean age 44.1 ± 11.4 years) without any voice complaints or chronic medical/psychiatric conditions. The control participants were frequency-matched to the patients on age and sex. All participants were from the southeastern region of Turkey and shared a similar cultural background, minimizing sociocultural differences between the groups. The participants’ occupations were diverse, including some in voice-intensive jobs (e.g., teachers, salespersons) and others in non-vocal professions; we did not specifically select or exclude professional voice users in either group. There were no statistically significant differences between the hoarseness and control groups in terms of the mean age or sex distribution (*p* > 0.05). Detailed demographic characteristics of the participants are presented in Table 1. Based on clinical evaluations by otolaryngologists, the etiologies of hoarseness in the patient group were heterogeneous. The majority of patients had identifiable organic lesions or conditions (such as vocal fold nodules, polyps, chronic laryngitis, or other structural/neurological causes of dysphonia), while a smaller subset had functional dysphonia (voice disturbance without a visible lesion). Due to the limited sample size, patients were not subdivided by diagnosis for statistical analysis.

### 2.3. Data Collection

All participants completed a sociodemographic and medical history form, along with the validated Turkish versions of the quality of life and anxiety questionnaires described below. After providing informed consent, participants were assessed in person. For patients, the questionnaires were administered during their clinic visit (after their evaluation by the ear, nose, and throat (ENT) physician for hoarseness). Healthy controls were recruited from the community (including hospital staff and local volunteers) via announcements and word-of-mouth, and they completed the assessments either at the hospital or in a quiet setting such as their home or workplace, under the guidance of the research team. Participants first filled out the sociodemographic/medical information form and then proceeded to the self-report questionnaires. The questionnaires were self-administered (paper-and-pencil format) in a fixed order for all participants: first the WHOQOL-BREF, followed by the LSAS, and then the HADS. A member of the research team was present or readily available to answer any questions and ensure standardized administration. The average time to complete all questionnaires was approximately 20–30 min.

### 2.4. WHOQOL-BREF

The World Health Organization Quality of Life Brief Form (WHOQOL-BREF) is a general quality of life instrument with 26 items that assess individuals’ perceptions of their quality of life over the past two weeks, yielding four domain scores: Physical Health, Psychological Health, Social Relationships, and Environmental Conditions ([19]). The Turkish version of the WHOQOL-BREF has been validated and is scored such that each domain yields a score from 0 to 100, with higher scores indicating a better quality of life. In our study, we calculated domain scores for each participant according to standard scoring guidelines.

### 2.5. Liebowitz Social Anxiety Scale (LSAS)

The LSAS is a 24-item self-report scale used to measure the severity of social anxiety disorder. The scale assesses an individual’s fear (anxiety) and avoidance in a variety of social situations, comprising 11 social interaction situations and 13 performance situations ([11]). Each situation is rated on separate 0–3 scales for fear and avoidance, and the instrument provides a total social anxiety score as well as two context-specific subscale scores: one for Social Interaction Anxiety (everyday interpersonal situations) and one for Performance Anxiety (situations involving performance or being observed). The officially validated Turkish version of the LSAS was used in this study ([21]).

### 2.6. Hospital Anxiety and Depression Scale (HADS)

The HADS is a 14-item self-administered questionnaire measuring anxiety and depressive symptom levels over the past week. The scale, for which a Turkish validation study is available, consists of two subscales: Anxiety (HADS-A) and Depression (HADS-D) ([26]). Scores on each subscale range from 0 to 21, with higher scores indicating greater symptom severity. By conventional cutoff values, scores are interpreted as 0–7 = normal, 8–10 = borderline (mild), and ≥11 = clinical range for anxiety or depression ([26]). In this study, HADS anxiety and depression scores were calculated for each participant and used for between-group comparisons.

### 2.7. Statistical Analysis

Statistical analysis was performed using SPSS version 26.0. Descriptive statistics were used to summarize sample characteristics. Continuous variables (questionnaire scores) were first examined for distribution normality. Group comparisons between the hoarseness and control groups were conducted using appropriate tests: independent-sample *t*-tests for approximately normally distributed continuous variables (or Mann–Whitney U tests for non-normal distributions) and chi-square tests for categorical variables (e.g., sex distribution). A significance level of *p* < 0.05 (two-tailed) was used for all hypothesis tests. In addition to *p*-values, we calculated standardized effect sizes for the main between-group differences. For the *t*-test results, Cohen’s d was computed to estimate the magnitude of group differences (with benchmarks of approximately 0.2 for small, 0.5 for medium, and 0.8 for large effects). We also report 95% confidence intervals for the mean differences or effect sizes where relevant, to indicate the precision of the estimates. No formal a priori sample size calculation was performed for this observational study; however, a post hoc power analysis indicated that our sample (38 patients and 40 controls) provides about 80% power to detect between-group differences of moderate-to-large effect sizes (approximately d ≥ 0.7) at α = 0.05.

### 2.8. Psychometric Properties

We assessed the internal consistency of each scale in our sample. All instruments demonstrated good reliability. For the LSAS, Cronbach’s alpha was 0.95 for the total score, with α = 0.93 for the Social Interaction subscale and α = 0.91 for the Performance subscale, indicating excellent internal consistency. The HADS showed solid reliability as well (Cronbach’s α = 0.84 for the Anxiety subscale and 0.78 for the Depression subscale). The WHOQOL-BREF domain scores had acceptable internal consistencies given the small number of items per domain: Cronbach’s α coefficients were 0.82 for Physical Health, 0.80 for Psychological Health, 0.65 for Social Relationships, and 0.79 for Environmental Conditions. These values are comparable to those reported in validation studies of the Turkish WHOQOL-BREF (e.g., most domain α > 0.70, with the Social domain often slightly lower due to having only three items). Overall, the scales used in this study were found to be psychometrically robust for our sample, supporting the reliability of the findings.

## 3. Results

### 3.1. Participant Characteristics

After exclusions, the final sample included 38 patients with hoarseness and 40 healthy controls. The two groups were closely matched on basic demographics: the hoarseness group’s mean age was 45.3 (SD = 12.1) and the control group’s mean age was 44.1 (SD = 11.4), and the sex distributions (approximately equal numbers of females and males in each group) were similar. There were no significant differences between the groups in age (*p* = 0.68) or sex ratio (*p* = 0.81). Table 1 summarizes the demographic characteristics. Aside from the presence of a voice disorder in the patient group, the groups were comparable in other recorded variables.

### 3.2. Quality of Life Outcomes (WHOQOL-BREF)

Group mean scores for the Physical Health, Psychological Health, Social Relationships, and Environmental domains of the WHOQOL-BREF for both groups are shown in Table 2. Patients with hoarseness had significantly lower scores than the controls on the Psychological Health and Social Relationships domains (*p* < 0.05 for both), indicating a worse quality of life in these areas. There were no significant group differences in the Physical Health or Environmental domain scores (*p* > 0.05). This pattern suggests that while overall physical well-being and satisfaction with the environment were similar between the two groups, the hoarseness group experienced notable deficits in emotional well-being and social life satisfaction relative to healthy controls.

### 3.3. Social Anxiety Outcomes (LSAS)

As shown in Table 3, patients with hoarseness exhibited higher social anxiety on the LSAS compared to the controls. The total LSAS score was significantly greater in the hoarseness group (mean 62.8 ± 21.4) than in the control group (47.5 ± 18.9, *p* = 0.003), reflecting greater overall social anxiety in patients. This difference corresponds to a large effect (Cohen’s d ≈ 0.76). In particular, the Social Interaction subscale score was markedly elevated in patients with hoarseness (34.1 ± 11.5) versus controls (27.8 ± 10.3, *p* = 0.021), with a medium effect size (d ≈ 0.58). In contrast, the Performance Anxiety subscale did not differ significantly between the groups (28.7 ± 10.9 vs. 25.1 ± 9.8, *p* = 0.17, d ≈ 0.35). Thus, individuals with hoarseness reported considerably more anxiety in routine social interaction situations (e.g., meeting new people, talking in small groups) but not in formal performance situations (e.g., speaking or performing in front of an audience). The LSAS explicitly differentiates between these two types of situations, and our results indicate that the elevated social anxiety associated with hoarseness is context-specific to day-to-day interpersonal interactions rather than generalizing to performance-based scenarios. This finding suggests that the social anxiety experienced by the hoarseness group is more pronounced in ordinary social encounters, aligning with the kinds of communicative situations that are frequently challenging for individuals with a hoarse voice.

### 3.4. General Anxiety and Depression Outcomes (HADS)

The group results for the Hospital Anxiety and Depression Scale are presented in Table 4. The mean anxiety score (HADS-A) in the patient group was 8.5 ± 3.7, compared to 7.9 ± 3.4 in the control group; this difference was not statistically significant (*p* = 0.52) and was very small in terms of the effect size (Cohen’s d ≈ 0.17). Similarly, the mean depression score (HADS-D) was 6.7 ± 3.1 in the patients and 6.1 ± 2.8 in the controls (*p* = 0.47, d ≈ 0.20), indicating no significant difference between the groups. Notably, in both groups, the average HADS scores fell into the “borderline” range (8–10) rather than the “normal” range (0–7) for anxiety and were in the upper end of the normal range or mild range for depression. In other words, even the healthy control group reported a mild level of anxiety and depressive symptoms on average. The absence of a significant group difference on the HADS suggests that chronic hoarseness was not associated with any marked increase in general anxiety or depression levels beyond this mild baseline. This finding reinforces that the psychological impact of hoarseness observed in our sample was relatively specific to social anxiety and did not extend to generalized anxiety or mood disorder symptoms in a statistically detectable way.

## 4. Discussion

This case–control study provides a comprehensive examination of the quality of life and anxiety profiles of individuals with hoarseness, and it yields several important findings. Consistent with our hypotheses, chronic hoarseness was found to adversely affect certain domains of quality of life and to be associated with elevated social anxiety. Specifically, psychological health and social relationship quality of life scores (WHOQOL-BREF domains) were significantly lower in patients with dysphonia than in healthy controls. This suggests that individuals with voice disorders experience diminished emotional well-being and satisfaction with their social life compared to those without voice problems. No significant group differences were observed in the physical health or environmental domains of quality of life. The Physical Health domain encompasses facets such as energy, pain, and the ability to perform daily activities, while the Environmental domain assesses factors like financial resources, physical safety, and access to services. Our findings imply that individuals with hoarseness feel as physically healthy and materially secure as their healthy peers, yet they face pronounced difficulties in psychological and social aspects of life. This observation supports the notion that voice disorders impose a communicative and emotional burden rather than directly affecting one’s perceived physical health or external environment ([7]). Similarly, a study by [12] ([12]) demonstrated a markedly reduced quality of life in patients with dysphonia, an effect that was independent of the type of voice disorder (organic vs. functional) or the patient’s sex. Those authors also noted that the degree of the quality of life impact might vary across different cultural populations, cautioning that the effects of dysphonia on one’s quality of life may not be uniform in every society ([12]). In our sample drawn from Turkey, the finding that primarily the psychological and social domains of one’s quality of life are affected suggests that the importance attributed to voice in our cultural context manifests chiefly in those domains.

Our study also showed that individuals with hoarseness have significantly higher levels of social anxiety compared to healthy controls ([14]). This difference was most pronounced in everyday social interaction situations, whereas no marked difference was observed in special performance-related situations. Patients with hoarseness had substantially elevated total scores on the Liebowitz Social Anxiety Scale relative to the controls. In particular, anxiety levels in scenarios requiring routine social interaction (e.g., conversing with strangers, speaking in a small group, dining in public) were higher in the patients, whereas in performance situations such as giving a speech or being the center of attention in a formal setting, the patients and controls reported similar levels of anxiety. This pattern suggests that the social anxiety experienced by individuals with hoarseness is more specific to day-to-day communication contexts and does not generalize as strongly to formal performance situations. Notably, this conclusion is directly supported by the LSAS subscale data: only the Social Interaction subscore (covering everyday interactions) was significantly elevated in our patient group, while the Performance Anxiety subscore was not. In other words, patients’ anxiety was selectively heightened in casual, interpersonal settings, but not during performance or audience-based scenarios. Clinically, this finding makes intuitive sense—someone with a chronically hoarse voice might worry about one-on-one or small group conversations (where being misunderstood or drawing attention due to an abnormal voice could cause embarrassment), yet they might not exhibit disproportionately high anxiety for public performances if such situations are equally intimidating for everyone or if they avoid those altogether. Our result here aligns with the idea that the psychosocial impact of hoarseness manifests predominantly in social functioning.

In the literature, certain types of voice disorders (especially functional dysphonias) have been associated with higher levels of anxiety and distinct personality traits ([15]). For example, patients with functional (non-organic) dysphonia have been found to exhibit elevated anxiety and depression scores compared to population norms ([25]), and a study by Willinger et al. reported that about one-third of patients with functional dysphonia had psychiatric symptoms meeting criteria for major depression or an anxiety disorder. In our study, although social anxiety symptoms were prominent in the hoarseness group, general anxiety and depression levels (as measured by the HADS) were not significantly elevated compared to the controls. This is likely attributable to the predominance of organic voice disorders in our patient sample, rather than purely functional ones ([8]). As noted in the Methods, most patients’ hoarseness stemmed from structural or organic etiologies (such as nodules, polyps, etc.), which might inherently cause communication difficulty, but not be as strongly tied to underlying psychopathology as functional dysphonia can be. In certain subgroups of dysphonia—for instance, spasmodic dysphonia, which causes severe involuntary voice breaks and can be very disabling—higher rates of psychiatric comorbidity (especially social anxiety and depressive symptoms) have been reported ([10]). However, “hoarseness” as a general condition encompasses a heterogeneous group of causes, and the degree of psychosocial impact may vary with different etiologies. Recent evidence suggests that psychological distress is elevated in patients with voice disorders across the board, with only minimal differences between functional and organic diagnoses in terms of anxiety and depression levels. This meta-analytic finding indicates that even patients with organic voice disorders can experience considerable psychological effects ([3]). Our results are consistent with this view: despite most of our patients having organic causes of dysphonia, they still showed significant psychosocial challenges (specifically, heightened social anxiety), underscoring that the impact of a voice disorder is not confined to purely functional cases.

In our study, the lack of differences between patients and controls in general anxiety and depression (HADS scores) suggests no strong association between chronic hoarseness and non-specific anxiety or mood symptoms. Neither group had mean HADS scores in the purely “normal” range—both averages were in the mild or borderline elevated range for anxiety and in the normal-to-mild range for depression. In fact, our healthy control group, despite having no diagnosed health issues, reported a mild level of anxiety/depressive symptoms on average. This likely reflects that subclinical anxiety and depression are relatively common even in otherwise healthy individuals (e.g., due to everyday stressors or general life circumstances). Therefore, the hoarseness group’s similar mild levels of HADS anxiety/depression indicate that the presence of a voice disorder did not add significantly to the generalized psychological distress beyond this baseline. Stated differently, the psychosocial impact of hoarseness in our sample was specific (focused on social anxiety and related quality-of-life domains) rather than broad (extending to overall anxiety or depressive illness). Although patients with hoarseness exhibited notable social anxiety symptoms, their mean LSAS scores did not typically reach the threshold for a clinical diagnosis of social phobia. The average LSAS total score for the hoarseness group (around 63) is below commonly used clinical cutoffs for social anxiety disorder. This suggests that the social anxiety related to hoarseness, while clearly elevated, might be viewed more as a heightened situational fear or social discomfort rather than a full-blown anxiety disorder in many cases. Similarly, studies on healthcare workers during the COVID-19 pandemic have reported increased anxiety and depressive symptoms, though these did not always amount to clinical disorders ([4]). Therefore, the presence of situation-specific anxiety in patients with hoarseness can emerge without necessarily escalating to a generalized or clinically diagnosable psychiatric condition.

Despite the strengths of this study (such as the use of validated instruments and a controlled comparison group), several limitations should be acknowledged. First, the sample size was relatively small, which may limit the generalizability of the findings and the statistical power to detect more subtle effects. With only 38 patients, our study was adequately powered to detect medium-to-large differences, but may have missed smaller differences in some measures. Second, the cross-sectional design of this study does not allow us to determine the causality or directionality of the relationship between hoarseness and psychosocial distress. We have assumed that voice problems can lead to anxiety or quality-of-life reductions, but it is equally plausible that individuals with higher anxiety might be more prone to voice strain or that a third factor (such as a stressful lifestyle) contributes to both. Longitudinal data would be needed to untangle the cause and effect. Third, we did not perform a detailed subgroup analysis by type or cause of voice disorder. Our “hoarseness” group likely included a mix of conditions (e.g., nodules, polyps, functional dysphonia, etc.) and we treated them as a single group for analysis. It remains possible that certain etiologies of dysphonia have a greater psychosocial impact than others; for example, patients with functional dysphonia or neurological voice disorders might differ in their psychological profile from those with mild structural lesions ([18]). Due to the modest sample size, we were unable to compare such subgroups, which is a limitation in terms of the clinical specificity. Fourth, we did not systematically control for some potentially important confounding variables, such as participants’ occupation or extent of professional voice use, socioeconomic status, or other personality traits. Occupational voice demands in particular could influence both the likelihood of developing hoarseness and the degree of handicap it causes; for instance, a school teacher with hoarseness might experience more social and work-related consequences than a person who is retired or has a primarily non-verbal job. In our sample, we noted that participants came from varied occupations and we did not observe an obvious imbalance (e.g., both groups included a few teachers and voice professionals), but we did not formally measure or match the groups on this factor. Future studies should consider controlling for or stratifying by such factors. Finally, our participants were all drawn from a specific region and cultural context (Turkey) and the findings may not generalize to populations with different cultural attitudes toward voice disorders or different healthcare contexts. Cultural norms can influence how much a voice problem is stigmatized or how willing individuals are to report anxiety and depression, which in turn could affect the results. While we attempted to minimize sociocultural differences by drawing controls from the same community as the patients, caution is warranted in extrapolating these findings universally.

Looking ahead, there are several avenues for future research to build on these results. Larger studies with more participants would help confirm the observed differences in quality of life and social anxiety and would allow for more fine-grained analysis (such as comparing different types of dysphonia or conducting multivariate analyses to identify predictors of a psychosocial impact). It would be especially informative to compare functional versus organic voice disorder patients in terms of social anxiety and other psychological measures to see if the trends noted in this study and others ([2]) hold true in a more powered, direct comparison. Longitudinal studies are also needed to clarify the causal relationships: for example, following patients over time as they develop voice problems to see if anxiety levels increase, or conversely, treating the voice disorder (e.g., with surgery or voice therapy) to see if quality of life and anxiety improve subsequently. Such studies could help determine whether reducing hoarseness leads to reductions in social anxiety, or whether individuals with high social anxiety are at greater risk of voice strain and thus might need targeted preventive interventions. Additionally, interventional research could explore the benefits of integrating psychosocial care into voice disorder management ([23]). For instance, a randomized trial could test whether adding a short cognitive–behavioral therapy program focused on social anxiety (or other supportive counseling) for patients with dysphonia leads to better overall outcomes (voice-related handicap, quality of life, and mental health) compared to standard voice treatment alone. Beyond individual studies, it would be valuable to conduct cross-cultural research on voice disorders and psychosocial effects, as differences in social expectations and communication styles across cultures might modulate the impact of hoarseness on a person’s life.

Our findings, in conjunction with growing evidence in the field, underscore that patients with voice disorders often have significant psychological needs alongside their voice symptoms. A recent meta-analytic review confirmed that individuals with dysphonia, as a group, show elevated levels of depression and anxiety, and concluded that many of these patients could benefit from clinical psychology interventions ([2]). In light of this, it is important for clinicians to adopt a more holistic, multidisciplinary approach when managing patients with chronic hoarseness. Incorporating psychosocial considerations into the evaluation and treatment process—for example, routinely screening for social anxiety or depression in patients who present with voice problems and providing referrals to mental health professionals when appropriate—may improve treatment outcomes and the overall quality of care for these patients. Collaboration between otolaryngologists, speech–language pathologists, and psychologists or counselors can ensure that both the physical and emotional aspects of dysphonia are addressed. Ultimately, addressing the person behind the voice—including their mental well-being and social functioning—is likely to enhance the effectiveness of voice disorder management and help patients regain their confidence in communication.

## 5. Conclusions

Chronic hoarseness, beyond being merely a communication problem, can lead to significant consequences for an individual’s psychosocial well-being. In this case–control study, patients with hoarseness had a markedly lower quality of life, particularly in the psychological and social domains, compared to the healthy controls, while their perceived quality of life related to physical health and environmental conditions did not differ from that of the controls. Additionally, individuals with hoarseness exhibited significantly higher levels of social anxiety than the controls—especially in ordinary social interaction situations—whereas no group differences were found in general anxiety or depressive symptoms. Taken together, these findings indicate that hoarseness impacts mental health and social functioning in specific ways (notably by increasing situational social anxiety and reducing social/emotional quality of life) rather than manifesting as a generalized anxiety or mood disorder. From a practical standpoint, these results highlight the importance of looking beyond the vocal symptoms when treating dysphonia: healthcare providers should be aware of the potential social anxiety and emotional distress accompanying voice disorders. Interventions aimed at improving voice function may be enhanced by simultaneously providing support for these psychosocial issues. In particular, patients with hoarseness might benefit from counseling or therapy to manage social anxiety, alongside standard voice treatments. Future research should explore the efficacy of such combined approaches and investigate long-term outcomes, including whether alleviating social anxiety can improve patients’ overall quality of life and vocal rehabilitation results. By acknowledging and addressing both the voice and the psyche, clinicians can better help patients with hoarseness lead fuller, more socially engaged lives.

## Figures and Tables

**Table 1 behavsci-15-01160-t001:** Participant demographic characteristics.

Characteristic	Hoarseness (*n* = 38)	Control (*n* = 40)	*p*-Value
Age, years (mean ± SD)	45.3 ± 12.1	44.1 ± 11.4	0.68
Sex, female/male	20/18	22/18	0.81

SD: standard deviation.

**Table 2 behavsci-15-01160-t002:** WHOQOL-BREF quality of life scores.

Domain	Hoarseness (*n* = 38)	Control (*n* = 40)	*p*-Value
Physical Health	58.9 ± 11.6	62.5 ± 10.9	0.26
Psychological Health	52.4 ± 10.7	60.1 ± 9.3	0.004 **
Social Relationships	55.3 ± 12.5	63.8 ± 10.4	0.018 **
Environmental	61.0 ± 13.3	64.5 ± 12.1	0.34

Note: WHOQOL-BREF = World Health Organization Quality of Life—Brief Form. ** *p* < 0.01 indicate statistically significant differences.

**Table 3 behavsci-15-01160-t003:** Liebowitz Social Anxiety Scale (LSAS) results.

Subscale	Hoarseness (*n* = 38)	Control (*n* = 40)	*p*-Value
Social Interaction	34.1 ± 11.5	27.8 ± 10.3	0.021 **
Performance Anxiety	28.7 ± 10.9	25.1 ± 9.8	0.17
Total Score	62.8 ± 21.4	47.5 ± 18.9	0.003 **

Note: LSAS = Liebowitz Social Anxiety Scale. Higher scores indicate greater social anxiety. The Social Interaction subscale covers anxiety in everyday social encounters, whereas the Performance subscale covers anxiety in performance or observation situations. ** *p* < 0.01 indicate statistically significant differences.

**Table 4 behavsci-15-01160-t004:** Hospital Anxiety and Depression Scale (HADS) results.

Subscale	Hoarseness (*n* = 38)	Control (*n* = 40)	*p*-Value
Anxiety (HADS-A)	8.5 ± 3.7	7.9 ± 3.4	0.52
Depression (HADS-D)	6.7 ± 3.1	6.1 ± 2.8	0.47

Note: HADS-A = Hospital Anxiety and Depression Scale Anxiety subscale; HADS-D = Depression subscale. All *p*-values > 0.05 (no statistically significant differences). According to standard interpretation, mean scores in both groups fell in the mild (“borderline”) range for anxiety (8–10) and in the normal-to-mild range for depression.

## Data Availability

The data presented in this study are available on request from the corresponding author.

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
