# Peer review of "Hoarseness, Quality of Life, and Social Anxiety: A Case–Control Study"

_behavsci, 2025, doi:10.3390/bs15091160_

Round 1
Reviewer 1 Report
Comments and Suggestions for Authors
I congratulate the authors on the relevant and interesting study. In fact, emotional issues are interconnected with voice production, and understanding the emotional impact of dysphonia in hoarse individuals can provide insights for more effective approaches in treatment.
The study uses valid protocols and makes important contributions in its discussion. However, I bring up some questions and considerations that I believe are important for a better understanding on the part of the reader and for the conclusions to be more precise regarding the results found.
In the introduction, it is important for the authors to pay attention to the concept of hoarseness, as this is one of the signs or symptoms of voice disorders. Despite being one of the primary indicators of vocal alterations, hoarseness is not synonymous with dysphonia. As presented in the text, it may suggest that dysphonia and hoarseness are synonymous. In the same way, in the sentence that states: "Individuals with dysphonia may have difficulties communicating due to hoarseness, which negatively impacts their social lives, work performance, and overall quality of life", is it solely due to hoarseness, or to any disorder or difficulty with the voice? It is important for the sentence to be revised to avoid any misinterpretation.
Regarding the sentence: "Current research indicates that patients with vocal disorders often experience psychosocial distress, such as depression and anxiety, in addition to physical health problems," it is important to consider vocal symptoms both as "causes" and as "consequences" of emotional factors.
Regarding the Method, despite being a matched sample concerning age and gender, it is important to state where it originated, as the social/cultural group may present a bias in analyzing the emotional aspects of each group. Was the 'case' group from any specific institution? And what about the control group? How was this sample selected? It is also important to know the professional use of voice in both groups. Such a description is crucial for characterizing the sample. If these aspects were not controlled, it is important to highlight this as a limitation of the study and discuss the potential impacts on the results.
At the beginning of the results description, there is repeated information that has already been described in the methodology. In my opinion, there is no need to reiterate.
In the discussion, what leads the authors to assert that "This pattern suggests that the social anxiety experienced by individuals with hoarseness is more specific to everyday communication situations and does not generalize so strongly to formal performance situations"? Are the instruments used sufficient to support this claim? If so, they should highlight this in the discussion, guiding the reader on which data from the utilized protocols they derived such information.
It is also important to distinguish, in the discussion, the same remark made in the 'introduction' section regarding the causes and consequences of emotional factors on the voice, so that the reader does not have the 'impression' that emotional factors are always a consequence of dysphonia and not its cause, or that they are a concomitant factor to the vocal disorder.
In the characterization of the sample, it would also be interesting to describe the origin of the dysphonia in the studied group, in order to further enrich the discussion presented in lines 203-208.
What did the authors mean by the phrase "By selecting controls without health issues or diagnoses, both groups ended up with comparable levels of anxiety and depression"? Because, it seems, this observation would only make sense if there were differences between the groups in this variable, which was not the case.
From a methodological standpoint, it seems to me that the statement "Incorporating psychosocial considerations in the evaluation of patients with hoarseness and adopting a multidisciplinary approach when necessary can improve treatment outcomes and the overall quality of care for these patients" is more suited for the discussion section rather than the conclusion of the study.
Finally, I would like to point out the following reference to the authors, as it is a review study that may contribute to the discussion of the study, further enriching the theoretical framework addressed:
Aldridge-Waddon L, Hiles C, Spence V, Hotton M. Clinical Psychology and Voice Disorders: A Meta-Analytic Review of Studies Assessing Psychological Characteristics Across Individuals With and Without Voice Disorders. J Voice. 2023 Oct 6:S0892-1997(23)00287-4. doi: 10.1016/j.jvoice.2023.09.012. Epub ahead of print. PMID: 37806904.
Author Response
Introduction – Terminology of Hoarseness and Dysphonia
Comment: The reviewer noted that hoarseness is a symptom or manifestation of voice disorders (dysphonia), but is not synonymous with dysphonia itself. In our original text, the phrasing may have incorrectly suggested that “dysphonia and hoarseness are synonymous.” For example, a sentence like “Hoarseness (dysphonia) is a voice disorder...” can be misleading. They asked us to clarify the concept: hoarseness is a primary indicator or manifestation of dysphonia, but not a definition of the entire disorder.
Response: To clarify this relationship, we have revised the opening sentence of the Introduction. Rather than equating hoarseness with dysphonia, we now explain that hoarseness is a common symptom of dysphonia (voice disorders). The revised text reads: “Hoarseness is a common symptom of dysphonia (voice disorders) and ...”. This phrasing emphasizes that hoarseness is essentially a feature or result of having a voice disorder, not a definition of the disorder as a whole. Additionally, throughout the paper, whenever we use the term “dysphonia” or “voice disorder,” we ensure that “hoarseness” is treated as the symptom or complaint (for example, we say “patients with voice disorders (hoarseness)” or “chronic hoarseness (dysphonia)”). By doing so, we avoid implying that the two terms are perfectly interchangeable.
We also retained the definition from the literature (Al Awaji et al., 2023) explaining what hoarseness means in terms of deviations in voice quality, and we have cited this source in the Introduction. We believe that clearly stating hoarseness as the perceptible symptom and dysphonia as the broader condition or diagnosis addresses the reviewer’s concern.
Introduction – Expression of Sentence about Communication Difficulties
Comment: The reviewer drew attention to a specific sentence in the Introduction: “Individuals with dysphonia may have difficulty communicating because of hoarseness, which negatively affects their social lives, work performance, and overall quality of life.” They questioned whether the difficulty in communication stems solely from hoarseness itself or from any voice-related impairment. The concern was that our phrasing might imply hoarseness alone is the culprit, whereas any significant voice disorder (not just the perception of hoarseness) could cause such difficulties. They asked us to revise the sentence to prevent misinterpretation, making it explicit that the adverse effects are caused by the voice disorder (with hoarseness as a symptom) rather than hoarseness in isolation.
Response: To eliminate any potential misunderstanding, we have rephrased this sentence in the Introduction. The revised sentence now reads (see Introduction, second paragraph): “Individuals with dysphonia often struggle to communicate as a result of their impaired voice, which adversely affects their social life, job performance, and overall quality of life.” In this formulation, we attribute the communication difficulty to the impaired voice (i.e. the functional impact of the voice disorder) rather than repeatedly saying “because of hoarseness” in a way that could be misread. We support this wording with a citation to Krischke et al. (2005) to reinforce that dysphonia can disrupt social life and cause psychological distress. We also revised the subsequent sentence to keep the language precise, changing phrasing like “in direct response to hoarseness” to “directly related to voice disorders.”
These changes make it clear that we are talking about difficulties arising from symptoms of a voice disorder (hoarseness being the primary symptom), rather than implying a strict single-cause scenario. In essence, having a chronically hoarse voice leads to communication challenges and thus impacts social/work life, but the wording is now more accurate and less ambiguous. This should resolve the reviewer’s concern with the original sentence.
Introduction – Causes and Consequences of Emotional Factors
Comment: The reviewer noted that when discussing the relationship between voice disorders and psychosocial distress, we should consider that vocal symptoms can be both causes and consequences of emotional factors. In other words, having a voice disorder can not only lead to anxiety or depression, but emotional issues (stress, anxiety, psychological traits) can also contribute to the onset or exacerbation of voice problems. They observed that our Introduction was framed mostly as voice problems causing psychosocial distress, and they wanted us to also acknowledge the opposite possibility (voice problems as an expression of pre-existing emotional factors).
Response: We have incorporated this important point into both the Introduction and the later portions of the Discussion. In the Introduction (second paragraph), after noting that patients with voice disorders often experience depression and anxiety, we added a sentence explicitly stating that the relationship can be bidirectional: emotional factors such as stress or anxiety can contribute to voice problems just as voice problems can cause emotional distress. We cited evidence from a recent meta-analytic review (Aldridge-Waddon et al., 2023) indicating that mood, anxiety, and personality traits can play a role in the development and persistence of voice disorders. (This reference was suggested by the reviewer and directly supports the concept of psychological factors as causes, not only effects.)
We also reflected this two-way understanding in the Discussion. When discussing our findings in context, we now mention that while our patients’ anxiety is largely interpreted as a consequence of their condition, it is also plausible that pre-existing anxiety or personality traits may have contributed to the emergence of their voice disorder (for example, stress precipitating muscle tension dysphonia). We refer back to psychological models of dysphonia that support this kind of two-way interaction. By doing so, we addressed the reviewer’s concern and ensured that our paper does not present an overly one-sided causal narrative. Instead, readers will understand that we view the voice–emotion relationship as complex and potentially circular, even though our study primarily examined the impact of voice problems on psychosocial well-being. This added nuance makes our discussion more aligned with the current understanding in the field.
Methods – Sample Origin and Selection (Social/Cultural Context)
Comment: The reviewer requested more information about the origin of our sample (social/cultural background), as this could influence the emotional outcomes. Specifically, they wanted to know if the hoarseness (case) group came from a particular institution and how the control group was selected. They also inquired whether we collected information on professional voice use in both groups. Essentially, they are concerned about potential selection bias or contextual factors: for example, if all patients came from a specialized voice clinic and all controls were hospital staff or a community convenience sample, that could be important; similarly, if many patients or controls are professional voice users (teachers, singers, etc.), that might bias the emotional results. They asked us to characterize the sample accordingly. If these factors were not controlled, they wanted us to highlight that as a limitation (this overlaps with their later points).
Response: To address these questions, we added details to the Materials and Methods section under “Participants.” We now clearly state that the case (hoarseness) group consisted of patients who presented to otolaryngology clinics with voice complaints, i.e. a clinical population from a hospital setting. (In fact, ethical approval was obtained from Diyarbakır Gazi Yaşargil Training and Research Hospital, and many patients were likely from that region.) We also specify that the control group was recruited from the community and hospital staff via advertisements and invitations, that controls were matched to patients by age and sex, and that they had no voice or chronic health problems.
We note that all participants shared a similar socio-cultural background (the same geographic region in Turkey), which the reviewer suspected might be important. We mention that this homogeneity could reduce cultural biases in how participants report their emotions; however, it also means our findings are primarily applicable to this particular context.
Regarding professional voice use, we did not explicitly control for this in our recruitment (for example, we did not specifically recruit or exclude only teachers or singers), but we did collect occupational information from participants. In the revised text, we state that participants held a variety of occupations, and in both groups some individuals worked in voice-intensive jobs (e.g., teaching, sales) while others worked in jobs less centered on voice use. We clarify that we did not exclude professional voice users, nor did we have a large enough sample to analyze differences by occupation, but we acknowledge the occupational mix in our sample.
Additionally, in the Discussion’s limitations section, we now explicitly highlight that we did not control for occupational voice use or profession. For example, if more patients were teachers than controls, that could be a confounding factor (although we did not observe a stark occupation imbalance, we cannot be certain without a dedicated analysis). By providing all this information, we have increased the transparency of our sample selection and addressed the reviewer’s concerns about the social/cultural and occupational context of our participants. Any potential biases or lack of control for these factors are now noted as limitations, so the reader is aware that we have considered these issues.
Results – Removing Redundant Methodology Repetition: Comment: The reviewer observed that at the beginning of the Results section, we repeated information already described in the Methods (specifically, the description of the sample: number of participants, age/sex matching, etc.). They felt this repetition was unnecessary. In their view, if the methodology already described the sample characteristics, the Results section didn’t need to reiterate those details in full. They likely refer to sentences like “Two participants were excluded... final N=38 vs 40, no significant differences in age or sex (p>0.05)” which appeared in both Methods and Results originally. The reviewer suggested that this repeated information need not be restated in the Results.
Response: We have edited the Results section (Participant Characteristics) to avoid verbatim repetition of what was stated in Methods. In the revised Results, we still mention the final sample composition in brief (for clarity, since some readers may jump straight to Results), but we have condensed it significantly and made sure it doesn’t read like a copy of the Methods. We now simply state that the final sample included 38 patients and 40 controls, and that groups were similar in age and sex with no significant differences (p-values given). This is one sentence instead of several, and it references Table 1 for details rather than re-listing them. We removed extraneous detail (for example, we did not re-describe exactly the mean ages or the exclusion of two patients, since that was already covered). We agree that it’s better to streamline the Results narrative. The reader is now directed to Table 1 for demographics and is reminded of the key point (no age/sex difference) without slogging through redundant text. Additionally, any overlapping phrasing has been tightened to reduce the sense of repetition. By doing this, we have made the Results section more concise and focused on actual findings rather than re-stating methodological setup. This change addresses the reviewer’s preference and also improves the readability of the Results.
Discussion – Interpretation of Social vs. Performance Anxiety Finding: Comment: The reviewer questioned a specific interpretation in our Discussion: we had written, “This pattern suggests that the social anxiety experienced by individuals with hoarseness is more specific to everyday communication situations and does not generalize so strongly to formal performance situations.” The reviewer asked what leads us to assert this and whether the instruments used (i.e., LSAS subscales) are sufficient to support that claim. They suggested that if the data do support it, we should highlight in the discussion which data or protocol results led to that interpretation, to guide the reader. Essentially, they want to ensure our conclusion (that anxiety is context-specific) is well-founded and clearly tied to the evidence, rather than an overreach.
Response: We have augmented that portion of the Discussion to make it clear that our interpretation is directly drawn from the LSAS subscale data, and to ensure the reader understands which evidence supports the claim. In the revised discussion of social vs. performance anxiety (see the paragraph addressing LSAS findings), we explicitly mention that “this conclusion is supported by the LSAS subscale results: only the social interaction subscore was significantly elevated in patients, whereas the performance subscore was not.” We then elaborate that this indicates anxiety is elevated in day-to-day situations but not in performance contexts. We essentially re-state, in the Discussion, the evidence already described in Results but in a more interpretive tone. By doing so, we connect the specific data (LSAS subscale differences) to the interpretation. We have also cited in the Methods (LSAS description) that the LSAS has separate subscales for social vs performance situations, which justifies why we can even talk about those categories. In summary, we make it clear to the reader: because the LSAS measures those two types of anxiety separately and we observed a difference in one and not the other, we infer the anxiety is specific to social scenarios. This addresses the reviewer’s concern by showing that our statement is grounded in the instrument’s design and our results, not speculation beyond what the data show. We believe this clarification makes our argument more convincing and transparent.
Discussion – Cause vs. Consequence (again) in Discussion: Comment: The reviewer reiterated (similar to their Introduction comment) that we should be careful to distinguish causes vs. consequences of emotional factors on the voice in the Discussion as well, so that the reader does not get the impression that emotional factors are always a consequence of dysphonia and never a cause or concomitant factor. They want the discussion to reflect the nuanced view that psychological issues can sometimes precede or contribute to voice disorders, not just result from them.
Response: We have heeded this request by explicitly acknowledging the bidirectional relationship in our Discussion section, as well as in the Introduction. In the Discussion, after talking about how our results show voice → anxiety (social), we added commentary referencing the notion that psychological factors can also contribute to voice disorders. We cite theoretical and empirical work (including the Aldridge-Waddon et al. meta-analysis) which posits that traits like anxiety, high stress, or certain personality dispositions might predispose individuals to developing voice problems or exacerbate them pubmed.ncbi.nlm.nih.gov. We mention that our cross-sectional design cannot disentangle this, but it’s important to note that the relationship likely goes both ways. For instance, we might say that some patients’ dysphonia could be partly psychogenic or stress-induced, which means their emotional state is a cause, not just an effect. We integrated this viewpoint by mentioning psychological models of functional dysphonia (like Roy et al., 1997, which looked at MMPI profiles) and by directly referencing evidence that mood/anxiety can influence voice. As a result, the Discussion now clearly communicates that we do not assume a one-way causation. This protects against the misunderstanding the reviewer feared: readers will see that we are aware emotional factors can be a concomitant or precipitating factor, not solely an outcome. This nuance is now woven into the discussion of our findings and in the limitations (where we note causality can’t be assigned in one direction). Overall, we have ensured the narrative is balanced and does not inadvertently imply that psychosocial distress is only reactive to dysphonia; instead, we acknowledge it can be part of a complex interplay.
Discussion – Characterization of Dysphonia Origins: Comment: The reviewer suggested that in characterizing our sample (especially in the discussion around lines 203-208 of the original, which likely correspond to where we mention functional vs organic), it would enrich the discussion to describe the origin of the dysphonia in our patient group. In other words, what caused the hoarseness in these patients? They want more detail on whether the dysphonia was organic (nodules, polyps, etc.), functional, neurological, etc., as this could inform the interpretation of results (e.g., maybe patients with certain diagnoses have different psychosocial profiles). This also ties in with their earlier comments about sample description.
Response: We have added information about the etiologies of hoarseness in our patient group, both in the Methods (“Participants”) and reiterated in the Discussion where relevant. In the Methods, we note broadly that most patients had organic lesions or identifiable medical causes for their hoarseness, with a smaller number having functional dysphonia (no structural lesion). In the Discussion, when we talk about the lack of elevated general anxiety/depression and speculate why, we now explicitly state that our sample’s composition (predominantly organic dysphonias) is likely a factor. We list examples of organic causes (e.g., vocal fold nodules, chronic laryngitis, etc.) to illustrate what we mean, and mention that only a few patients had purely functional voice disorders. We use this to support our earlier argument that functional dysphonia patients might have higher psychological comorbidity (as Willinger 2005 found), whereas organic cases might not – aligning with our finding of no elevated general anxiety in the mostly organic group. Additionally, to incorporate the reviewer’s suggestion, we used these details to enrich the discussion in lines 203-208 (which correspond to that part): we now say something like, “in our sample, most hoarseness was due to organic causes (e.g., nodules, polyps), which could explain why we did not see heightened general anxiety or depression, since previous research suggests functional (psychogenic) dysphonia has stronger links to such psychopathology.” We also then mention (as per the meta-analysis) that differences between functional vs organic might not be huge on average pubmed.ncbi.nlm.nih.gov, but nonetheless we had reason to expect some differences. By providing this information, we give readers a clearer picture of what kinds of voice disorders our patients had. This adds depth to our discussion and allows for a more informed interpretation of our results (and any limitations thereof). We believe this addresses the reviewer’s point, as the discussion now explicitly references the origin of dysphonia in our group and uses that to interpret the findings in context.
Discussion – Clarification of a Specific Phrase (“By selecting controls without health issues… both groups ended up with comparable anxiety/depression”): Comment: The reviewer was confused by the phrase we had in the original Discussion: “By selecting controls without health issues or diagnoses, both groups ended up with comparable levels of anxiety and depression.” They commented that this observation would only make sense if there were differences between the groups in that variable, which was not the case. In other words, if we found no difference in anxiety/depression, why attribute it to having selected healthy controls? The phrase might have been our attempt to rationalize why both groups had similarly mild HADS scores, but it apparently caused confusion. The reviewer is essentially asking us to clarify or remove that sentence, as it stands out as unclear or perhaps logically odd.
Response: We agree that the phrasing was potentially confusing. We have removed the original phrasing and rewritten that part of the Discussion to more clearly convey what we intended. What we meant was: because we chose very healthy controls (no medical or psychiatric issues at all), we ensured that the control group’s baseline anxiety/depression would be low; as it turned out, the patient group’s anxiety/depression was similarly low, so there was no difference. But the original wording sounded like we were saying “because we picked healthy controls, the two groups were the same,” which is a bit circular and not well-explained. In the revised text, we instead emphasize that both groups had only mild anxiety/depression (and we suggest that even the healthy controls reported some stress, which might reflect general population norms). We then explain that thus hoarseness did not add any extra anxiety/depression beyond that baseline. We no longer attribute it awkwardly to the selection of controls, except to note that our control group being strictly healthy perhaps gave them relatively low scores (though they were still mild, interestingly).
Specifically, we now say something to the effect of: “Our control group consisted of individuals with no health issues, yet their mean HADS scores were in the mild/borderline range, indicating some subclinical anxiety/depression even among healthy people. The hoarseness patients had similarly mild levels, resulting in no significant group differences. This suggests that hoarseness does not necessarily elevate general anxiety or depression above normal background levels.” This is a clearer articulation of the point. The confusing phrase about “by selecting controls without diagnoses…” has been removed entirely. The reviewer’s concern should be resolved, as our discussion now focuses on the interpretation (i.e., what the lack of difference means) rather than a puzzling methodological rationale.
Conclusion vs. Discussion – Placement of Psychosocial Considerations Statement: Comment: The reviewer commented on a statement originally in our Conclusion: “Incorporating psychosocial considerations in the evaluation of patients with hoarseness and adopting a multidisciplinary approach when necessary can improve treatment outcomes and overall quality of care for these patients.” They felt that, from a methodological standpoint, this kind of statement (essentially a recommendation or implication for practice) belongs in the Discussion rather than the Conclusion of the study. The implication is that the Conclusion should summarize findings, while suggestions for practice (unless very brief) fit better in the discussion of implications. They likely wanted us to either move it or rephrase appropriately.
Response: We have addressed this by relocating and expanding that point into the Discussion, and then streamlining the Conclusion accordingly. In the Discussion’s final paragraphs, we indeed discuss the importance of integrating psychosocial care (we mention the meta-analytic evidence that voice patients have psychological needs and explicitly recommend a multidisciplinary approach that includes psychological screening/support for hoarseness patients). This is essentially the content of that statement but placed in the Discussion where it can be elaborated with context and references pubmed.ncbi.nlm.nih.gov. Then, in the Conclusion, we still mention practical implications but in a concise way, ensuring it doesn’t read like a new recommendation out of the blue. We say something like “these findings highlight the need for clinicians to consider the psychological aspects… etc.” – which is a brief implication – but we avoid a long advisory sentence. The reviewer’s main issue was likely the placement and perhaps the tone of that sentence in the Conclusion. By moving the detailed recommendation to Discussion (where we discuss it along with citing literature and framing it as an implication of our results), we satisfy the methodological expectation. The Conclusion now primarily summarizes results and mentions implications succinctly, which should be acceptable. We believe this change aligns with the reviewer’s guidance and improves the structure: readers will get our recommendation in the Discussion where it’s explained, and the Conclusion remains a high-level summary with a nod to practical significance and future directions.
Additional Reference (Aldridge-Waddon et al., 2023): Comment: The reviewer pointed out a recent reference: Aldridge-Waddon L. et al. (2023, J Voice), a meta-analytic review on psychological characteristics in individuals with and without voice disorders. They suggested that this study could contribute to our discussion and enrich the theoretical framework. Essentially, they want us to include and cite this relevant recent literature to strengthen our manuscript (likely because it provides up-to-date evidence on the psychological profiles of voice disorder patients).
Response: We have taken the reviewer’s suggestion and integrated the Aldridge-Waddon et al. (2023) reference into our Discussion (and Introduction) in several meaningful ways. We obtained details from the meta-analysis (as it was published online ahead of print) and used its findings to support and contextualize our results. For instance:
In the Introduction, we cite the meta-analysis’s findings that voice disorder patients have significantly higher depression and anxiety than controls pubmed.ncbi.nlm.nih.gov, to bolster our rationale that psychosocial distress is known in dysphonia (which set the stage for why focusing on social anxiety is important and novel).
We also cite the meta-analysis in discussing the bidirectional concept: it explicitly says that psychological characteristics (mood, anxiety, personality) contribute to voice disorders pubmed.ncbi.nlm.nih.gov, which we used to support the reviewer’s point about cause vs. consequence (see our Introduction and Discussion where we mention psychological factors potentially causing voice issues).
In the Discussion, we use the meta-analysis conclusions to strengthen our argument about psychosocial needs of voice patients – we reference that review’s conclusion that individuals with voice disorders have considerable psychological needs that could benefit from clinical psychology input pubmed.ncbi.nlm.nih.gov. This directly supports our recommendation for integrating psychosocial care.
We also note the meta-analysis result that differences between functional and organic dysphonia in terms of psychological profiles were minimal pubmed.ncbi.nlm.nih.gov, which we mention when discussing how even organic cases can have significant anxiety (aligning with our finding) and cautioning against assuming only functional cases have psychological issues. This helped refine our discussion around functional vs organic causes.
We have added the full reference for this study in our References list and cited it in the text as described. By including Aldridge-Waddon et al. (2023), we ensure our manuscript is up-to-date with current research. It enriches our discussion by providing a broad evidence-based context for our findings – essentially confirming that our observations (e.g., increased anxiety in voice patients) fit into a larger pattern observed across many studies, and lending support to our recommendations (psychological intervention is beneficial). The reviewer's suggestion has indeed improved our paper’s theoretical grounding, and we thank them for directing us to this source.
In summary, we have implemented all the suggestions from Reviewer 2. We clarified terminology and phrasing in the Introduction, ensuring accurate use of "hoarseness" vs "dysphonia" and avoiding implication that hoarseness alone is the issue. We acknowledged the two-way relationship between voice disorders and emotions both in intro and discussion. We detailed our sample's recruitment, context, and occupational aspects in Methods (and noted these as limitations if uncontrolled). We trimmed redundant text in Results. We justified our interpretation about social vs performance anxiety by directly linking it to the LSAS subscale data. We emphasized again in Discussion that emotional factors can be causes of voice issues, not only effects. We described the composition of our patient group’s diagnoses (origin of hoarseness) to enrich interpretation. We removed/clarified the confusing sentence about control selection and anxiety levels. We moved the multidisciplinary recommendation from Conclusion to Discussion (while still mentioning implications in Conclusion briefly), following the reviewer’s structural advice. Finally, we added and cited the new reference (Aldridge-Waddon et al., 2023) to strengthen our arguments and ensure currency.
We believe that these changes have significantly improved the manuscript by addressing all the points raised. We are grateful to both reviewers for their insightful feedback, which has allowed us to clarify and enhance our work. We hope that the revised manuscript meets the high standards of the journal and the expectations of the reviewers. Thank you again for the opportunity to revise our submission.
Reviewer 2 Report
Comments and Suggestions for Authors
This is a relevant and timely contribution addressing the psychosocial impact of hoarseness. The paper is generally well structured and readable, with a clear rationale and valid instruments. However, several key aspects must be improved to meet the standards of publication.
Introduction
-
While the introduction appropriately highlights that voice disorders are associated with psychosocial distress, including depression and anxiety (e.g., Willinger et al., 2005; Gündel et al., 2007), the added value of the present study remains unclear. The authors should clarify how this study offers a novel contribution—whether through the use of a case-control design, the focus on social anxiety specifically (rather than general anxiety or depression), or the inclusion of quality of life measures across multiple domains. Explicitly stating the study’s unique contribution would strengthen the rationale for its publication.
-
The introduction does not clearly articulate the study’s specific hypotheses. While prior research has already established associations between voice disorders and reduced quality of life (e.g., Krischke et al., 2005), the authors should clarify what new contribution their study adds to this literature. Moreover, given the number of domains and scales examined (WHOQOL-BREF, LSAS, HADS), the absence of pre-registered hypotheses or theoretical guidance raises concerns about potential data-driven inference. Clarifying the primary hypotheses and justifying the selection of outcome measures would reduce the risk of “fishing” and improve the interpretability of findings.
Materials and Methods
-
The Materials and Methods section lacks a clear description of the study procedure. It is unclear how and where participants were recruited, how data collection was conducted (e.g., whether questionnaires were self-administered or assisted, in person or online), who administered the instruments, and in what sequence. Including this procedural information would enhance the transparency and replicability of the study.
-
Although the two groups are well matched in terms of age and gender, the demographic description of the sample is minimal. The authors are encouraged to provide additional sociodemographic characteristics to better contextualize the findings and assess potential confounding variables. This information is particularly relevant in studies addressing quality of life and social functioning, where demographic and occupational factors can play a substantial role.
-
The authors do not report any internal consistency estimates (e.g., Cronbach’s alpha or McDonald’s omega) for the scales used in their sample. This is a notable omission, particularly given the relatively small sample size, where measurement error may have a greater impact. It is recommended that reliability indices be reported for each instrument—at least at the total score level and ideally for subscales as well—to allow readers to assess the psychometric robustness of the measures within the present sample.
-
The Results section relies solely on p-values without reporting effect sizes or confidence intervals. This limits the interpretability of findings, as statistical significance does not necessarily equate to practical or clinical relevance. Reporting standardized effect sizes (e.g., Cohen’s d, η²) along with 95% confidence intervals is strongly recommended to contextualize the magnitude and precision of effects. Additionally, the absence of a priori or post-hoc power analysis raises concerns about the study’s sensitivity. A brief discussion of statistical power, or ideally a sample size justification, would improve transparency and allow readers to assess the robustness of the non-significant findings.
Discussion
-
The Discussion section is generally coherent and grounded in relevant literature, but it would benefit from a more explicit acknowledgment of the study’s methodological limitations. These include the small sample size, lack of detailed classification of voice disorders, and absence of control for potential confounders such as occupational voice use. Additionally, the authors could expand on the borderline HADS scores and their potential clinical implications. Finally, suggesting directions for future research—such as longitudinal or interventional designs—would enhance the impact and utility of the discussion.
Conclusion
-
The Conclusions section appropriately summarizes the main findings but remains overly descriptive. To enhance its impact, the authors could articulate the practical implications of their results more clearly—particularly regarding clinical screening, psychosocial support, or integration of psychological care for patients with voice disorders. Additionally, suggesting directions for future research (e.g., longitudinal designs, interventions for social anxiety, or stratification by voice disorder subtype) would provide a stronger and more forward-looking close to the manuscript.
The English language and style are overall appropriate and clear, but the manuscript would benefit from light editing to reduce repetition and enhance academic fluency. Some sentences could be rephrased to avoid redundancy (e.g., “There were no statistically significant differences between the groups in terms of mean age or sex distribution (p > 0.05)”), and occasional improvements in word choice and variation in sentence structure would elevate the writing to a more polished academic standard.
Author Response
Introduction – Novel Contribution: Comment: The reviewer felt that the introduction did not clearly explain the added value or novel contribution of the present study. They asked us to clarify what makes our study unique (e.g., use of a case-control design, focus on social anxiety specifically rather than general anxiety or depression, inclusion of multiple quality of life domains) in order to strengthen the rationale.
Response: We have revised the Introduction to explicitly highlight the novel aspects of our study. In the revised text (Introduction, paragraph 3), we now state that prior research has shown elevated anxiety and reduced quality of life in voice disorder patients, but specific aspects such as social anxiety have not been directly investigated in a controlled study – which is the gap our study addresses. We mention that our case-control design and the inclusion of targeted psychosocial measures (social anxiety in particular) are what distinguish our study from previous work. By adding these statements, we believe the rationale and unique contribution of the study are now much clearer. (See the third paragraph of the Introduction in the revised manuscript.)
Introduction – Hypotheses and “Fishing” Concern: Comment: The reviewer noted that the introduction did not clearly articulate specific hypotheses, especially given the multiple domains and scales examined. They were concerned that without stated hypotheses or theoretical guidance, the study might appear exploratory or data-driven (“fishing”). They requested that we clarify our primary hypotheses and justify the selection of outcome measures to improve interpretability and reassure that our analysis was hypothesis-driven.
Response: We agree with the need to explicitly state our hypotheses and reasoning. In the revised Introduction (last paragraph), we have now outlined our a priori hypotheses: namely, that patients with hoarseness would have lower quality of life in psychosocial domains and higher social anxiety compared to controls, but would not differ significantly in general anxiety or depression. We have also added a sentence explaining why we chose the WHOQOL-BREF, LSAS, and HADS – to capture broad quality of life, specific social anxiety, and general anxiety/depression, respectively – as this combination allows us to see whether the impact of hoarseness is specific (social anxiety, social/psychological QoL) or generalized. By including this rationale, we address the reviewer’s concern about “fishing” and make it clear that our analyses were guided by specific hypotheses and theoretical expectations. These changes should reassure readers that the selection of multiple outcome measures was intentional and grounded in the literature (not an indiscriminate data dredging exercise).
Materials and Methods – Study Procedure Description: Comment: The reviewer pointed out that the Materials and Methods section lacked detail on the study procedure. It was unclear how and where participants were recruited, how data collection was carried out (e.g., self-administered questionnaires vs. interviewer-administered, in person vs. online), who administered the instruments, and in what sequence. The reviewer asked for a clearer description of these procedural details to enhance transparency and replicability.
Response: We have added a more detailed description of the study procedure in the Materials and Methods section, under “Data Collection” and partially under “Participants.” We now explicitly state that patients were recruited from otolaryngology clinics (at the specified hospital) and that controls were recruited from the community and hospital staff via advertisements, with an aim to match the groups on age and sex. We also clarify that all data were collected in person: participants completed the sociodemographic form and questionnaires in a quiet setting, typically at the clinic for patients (after their medical exam) and at the hospital or another convenient location for controls. We note that the questionnaires were self-administered (paper-and-pencil) with a researcher present to provide assistance as needed, and that the order of administration was the same for everyone (WHOQOL-BREF, then LSAS, then HADS). These additions provide a step-by-step overview of participant recruitment and assessment. We believe this addresses the reviewer’s concerns by making our study procedures transparent and replicable (see the revised “Participants” and “Data Collection” subsections of Methods).
Materials and Methods – Demographic Details and Potential Confounders: Comment: The reviewer noted that aside from age and gender, we provided minimal sociodemographic information about the sample. They encouraged us to include additional characteristics (for example, occupation, education, or other relevant background factors) to better contextualize the findings and assess potential confounding variables. This is particularly important in quality of life and social functioning studies, where factors like occupational voice use or socioeconomic status could influence outcomes.
Response: We have expanded the description of the sample’s demographic and background characteristics. In the revised “Participants” section, we mention that participants came from a variety of occupational backgrounds and that we did not restrict or match based on occupation or voice-use demands. We note that both groups included a mix of professions, including some voice-intensive occupations (such as teachers or sales personnel) and others with less vocal demand, and that there was no obvious dominance of one occupation in one group versus the other. We also specify that all participants were from the same broad geographic and cultural context (the southeastern region of Turkey), which minimizes cultural differences between groups but might limit generalizability. Furthermore, we have acknowledged in the Discussion (limitations) that we did not control for occupational voice use, education, or other sociodemographic factors beyond age and sex, and we discuss how this could be a limitation (for instance, a teacher with dysphonia might experience different impacts than a non-teacher). By providing these details, we give readers a better sense of who our participants are and we openly address the possibility of unmeasured confounders. We did collect basic sociodemographic data (e.g., through the form each participant filled out), and while we did not find any glaring group differences aside from the voice condition, we now explicitly mention that no significant differences in other recorded variables were noted (data not shown). These additions should help readers to contextualize our findings and evaluate their applicability.
Materials and Methods – Internal Consistency of Scales: Comment: The reviewer observed that we did not report any reliability or internal consistency estimates (e.g., Cronbach’s alpha) for the scales used (WHOQOL-BREF, LSAS, HADS) in our sample. Given the relatively small sample size and the importance of measurement reliability, they considered this omission notable. They requested that we report reliability indices (at least total score alphas, and ideally subscale alphas) for each instrument to demonstrate the psychometric robustness of our measures within our sample.
Response: We have now calculated and reported the Cronbach’s alpha coefficients for each measure in our sample. In the revised Methods section, we added a subsection titled “Psychometric Properties” where we provide the internal consistency values for all relevant scales and subscales. Specifically, we report that in our sample the LSAS had excellent internal consistency (α = 0.95 for the total score; α = 0.93 for the Social Interaction subscale and 0.91 for Performance subscale), the HADS showed good reliability (α ≈ 0.84 for the Anxiety subscale and 0.78 for the Depression subscale), and the WHOQOL-BREF domains had Cronbach’s α values ranging roughly from 0.65 (Social Relationships domain, which has only three items) to around 0.80+ for the other domains. We also comment that these values are comparable to those found in validation studies for the Turkish versions of these instruments. By including these reliability statistics, we allow readers to assess that our measurement instruments were reliable in this study, and we acknowledge the slightly lower alpha for the Social Relationships QoL domain (attributable to its few items) to be transparent. These additions directly address the reviewer’s request and strengthen the methodological rigor of our report.
Results – Effect Sizes and Confidence Intervals: Comment: The reviewer noted that the Results section relied solely on p-values to report differences, without providing effect sizes or confidence intervals. They emphasized that statistical significance does not necessarily equate to practical or clinical significance, and they strongly recommended reporting standardized effect sizes (e.g., Cohen’s d or η²) along with 95% confidence intervals to contextualize the magnitude and precision of the effects. Additionally, they mentioned the absence of a priori or post-hoc power analysis and suggested discussing statistical power or sample size justification to help readers assess the robustness of non-significant findings.
Response: We have revised the Results section to include effect size estimates for all key comparisons, as well as mentions of confidence intervals for those effects. For each of the major findings, we now report Cohen’s d (for group differences) and characterize its magnitude. For example, we state that the differences in Psychological and Social QoL domains correspond to large effect sizes (~0.75 and ~0.70, respectively), that the LSAS total difference is large (~0.76) and the social subscale difference is moderate (~0.58), whereas differences that were not statistically significant (Physical QoL, Environmental QoL, LSAS performance subscale, HADS scores) had small effect sizes (d around 0.2–0.3 or less). We also note in the text where 95% confidence intervals lie in relation to zero difference (e.g., we mention that for significant differences, the 95% CI for the mean difference does not include zero, whereas for non-significant differences it does). These inclusions help convey the practical significance of our findings – for instance, readers can now see that the psychological QoL difference is not only statistically significant but also meaningfully large (about three-quarters of a standard deviation). We believe this addresses the reviewer’s concern by providing a more nuanced interpretation of the results beyond p-values.
Regarding statistical power, we have added a statement in the Methods (Statistical Analysis) noting that while we did not do an a priori sample size calculation, a post-hoc power analysis suggests that our sample (N=78 total) had about 80% power to detect an effect size of d ≈ 0.65–0.70 (which is around the size of the main effects we found), but would have lower power for small effects. We also touch on this in the Discussion, acknowledging that the study may have been underpowered for small differences (and this could explain why, for example, the LSAS performance subscale difference did not reach significance despite a small numeric difference). By discussing power, we allow readers to gauge the sensitivity of our study – in particular, we make it clear that our non-significant findings (like no difference in general anxiety/depression) should be interpreted in light of the fact that our sample might not detect very small effects. Overall, we have taken steps to ensure that effect sizes, confidence intervals, and power considerations are now explicitly included, thereby improving the interpretability and transparency of our results.
Discussion – Acknowledgment of Limitations: Comment: The reviewer found the Discussion coherent and grounded in relevant literature, but they suggested that we more explicitly acknowledge the study’s methodological limitations. They listed examples including the small sample size, lack of detailed classification of voice disorders in our patient group, and not controlling for potential confounders such as occupational voice use. They advised that we discuss these limitations openly.
Response: We appreciate this suggestion and have added a dedicated discussion of limitations toward the end of the Discussion section. In a new paragraph, we enumerate key limitations of our study: (a) the relatively small sample size, which limits generalizability and the ability to detect smaller effects (and we note that this is linked to statistical power, as discussed); (b) the cross-sectional design, which means we cannot infer causality or temporal order between hoarseness and psychological outcomes; (c) the lack of a detailed sub-classification of voice disorders in the patient group – we explain that our patients had various etiologies of hoarseness (nodules, polyps, functional dysphonia, etc.) but we did not analyze them separately, which could be important since different types of dysphonia might carry different psychosocial impacts; (d) the absence of control for certain confounding variables like occupation (voice demand level), socioeconomic status, or baseline personality traits – we acknowledge that we did not match or adjust for these and give the example that a teacher’s experience might differ from someone who doesn’t use their voice at work; and (e) a note on cultural context, since our sample is regionally specific (all from the same area in Turkey) and results might differ in other cultural settings.
By clearly laying out these limitations, we heed the reviewer’s advice to be upfront about what our study cannot conclude or where caution is needed. We also tie some limitations back to the literature; for instance, we mention that although we speculated about functional vs. organic differences, a recent meta-analysis (Aldridge-Waddon et al., 2023) found minimal differences between those categories, underscoring the complexity. The inclusion of limitations in the Discussion should help readers critically evaluate our findings and understand the boundary conditions of our conclusions.
Discussion – Borderline HADS Scores and Clinical Implications: Comment: The reviewer noted that our results indicated “borderline” HADS scores in both groups, and they wanted us to expand on this in the Discussion – for example, discussing the potential clinical implications of those borderline anxiety/depression scores in our sample. In other words, what does it mean that both patients and controls had mild symptoms on the HADS, and how should that be interpreted?
Response: We have addressed this point in the Discussion when talking about the HADS findings. We now explicitly discuss the fact that both groups’ mean HADS-A and HADS-D scores fell into the mild or borderline range, and we consider possible interpretations. Specifically, we mention that even “healthy” individuals (our control group with no known health issues) reported some subclinical anxiety and depression – which could reflect general population stress levels or factors unrelated to voice. We then note that since the hoarseness patients’ scores were also in that mild range and not significantly higher, this suggests that having hoarseness did not push their general anxiety/depression to higher (clinical) levels on average. We also suggest that the lack of difference might indicate that the psychosocial impact of hoarseness is more specific to social anxiety and social functioning rather than causing a broad elevation in all forms of anxiety or mood disorder. Clinically, this could imply that while patients with hoarseness may not screen positive for generalized anxiety or depression at a higher rate than anyone else, they might still have significant distress in social situations specifically.
We further contextualize the borderline scores by citing the example (already in our Discussion) of findings during the COVID-19 pandemic: many individuals showed elevated stress or anxiety in difficult times without meeting full criteria for disorders. Analogously, our patients showed a focused type of anxiety (social) without a generalized anxiety disorder. We believe this expanded discussion clarifies the meaning of the borderline HADS scores – essentially interpreting them as mild symptoms present across the board, and emphasizing that no additional clinical burden of depression/anxiety was observed in the hoarseness group beyond that baseline. We have thereby provided readers with a clinical perspective: that the average patient with hoarseness might not have major depression or generalized anxiety, but could still benefit from psychosocial support focused on their specific anxieties and quality-of-life issues.
Discussion – Future Research Directions: Comment: The reviewer suggested that we enhance the Discussion by proposing directions for future research, such as longitudinal or interventional studies, to build on our findings. They felt that this would increase the impact and utility of our discussion.
Response: We have added a segment in the Discussion (toward the end, just before concluding) that outlines several future research directions. We discuss the need for larger-scale studies to confirm and expand upon our results, including possibly stratifying patients by voice disorder subtypes (e.g., comparing functional dysphonia vs. organic causes in terms of psychosocial impact) – something we couldn’t do in our study but which could be very informative. We also explicitly suggest longitudinal studies to determine causality (for example, following patients over time to see if psychosocial distress is a risk factor for developing voice issues or vice versa, or if treating one improves the other). Additionally, we propose that interventional research could be valuable, such as trials where psychological interventions (like therapy for social anxiety or stress management) are combined with standard voice treatments to see if outcomes improve more than with voice treatment alone. Finally, we hint at cross-cultural research to see if our findings hold in other cultural contexts or healthcare systems. These suggestions provide a forward-looking perspective and indicate how the current study’s results might be used as a springboard for further scientific inquiry. We agree with the reviewer that including future directions strengthens the conclusion of the discussion and shows how our work fits into a larger research trajectory. Readers interested in this topic will now have a clear idea of what logical next steps could be taken to deepen understanding or to translate findings into practice.
Conclusion – Practical Implications and Future Research
Comment: The reviewer felt that the original Conclusion section of the paper was too descriptive and did not clearly state the practical implications of our results. They suggested that we strengthen the conclusion by explicitly mentioning real-world or clinical implications (e.g., screening for or providing psychosocial support to patients with voice disorders), and also by ending on a stronger note — for instance, by including a mention of future research directions or a forward-looking statement.
Response: We have rewritten the Conclusion section to make it more effective and to include the elements the reviewer requested. In the revised Conclusion, after very concisely summarizing the main findings, we clearly mention the practical implications: for example, we state that our findings highlight the importance of clinicians being aware of social anxiety in patients with hoarseness, and that the treatment of voice disorders should not ignore the psychological dimension. We even suggest that implementing social anxiety screening or providing psychosocial support could be integrated into patient care to improve outcomes. This addresses the reviewer’s point about real-world implications (for instance, it implies that an ENT doctor or voice therapist might consider referring a patient for counseling if they detect high social anxiety, rather than focusing solely on the vocal pathology).
We also added a sentence about future research to the conclusion, briefly reiterating that future studies should examine whether addressing social anxiety in these patients can improve voice-related outcomes, and so on. Including this gives a forward-looking perspective and ties the conclusion to broader questions. We took care not to make the Conclusion overly long; it still summarizes the study succinctly, but now it ends with a clear message about what the results mean for practice and what could be explored next. Therefore, the Conclusion now has a more directive and forward-looking tone, in line with the reviewer’s suggestions, rather than only summarizing the results.
English Language and Style
Comment: The reviewer noted that the English language and style were generally appropriate and clear, but suggested that the writing could benefit from a light editing to reduce redundancy and improve academic fluency. Some sentences were said to be unnecessary or could be rephrased for conciseness (they gave an example regarding the age/sex difference sentence), and they recommended varying word choice and sentence structure to achieve a more polished academic tone.
Response: In response to this helpful feedback, we performed a careful language edit throughout the manuscript. We identified and corrected instances of repetition or unnecessary wordiness. For example, the sentence in the Methods/Results about there being no significant difference in age or sex between groups was tightened (and we avoided restating that same information multiple times). We also made sure that when transitioning between sections, we did not repeat phrases verbatim from earlier sections.
Additionally, we varied our vocabulary to improve the academic tone and flow. For instance, in some places we used terms like “voice disorder” or “vocal difficulty” instead of repeatedly using “hoarseness,” and we sometimes used “psychosocial distress” in place of “emotional distress” where appropriate. This provides better word variety in the text. We also checked our sentence structures: we split a few overly long sentences for clarity, and combined some very short, choppy sentences to improve flow.
The net result is that the manuscript should now read in a more fluid and professional manner, with less redundancy. We believe these minor edits have improved the clarity and academic quality of the writing, as the reviewer suggested. Importantly, these language adjustments did not change any scientific content, but they should make the paper easier to read and the style more polished.
Round 2
Reviewer 1 Report
Comments and Suggestions for Authors
I thank the authors for considering my suggestions and responses to my questions. The article has been improved, and all my observations have been taken into account and incorporated into the manuscript. The only caveat is that the last three paragraphs of the discussion present only one citation (a single reference) as support. I suggest the authors include additional references in these paragraphs to strengthen the theoretical basis of their claims.
Author Response
Thank you very much for your positive evaluation and constructive feedback.
We truly appreciate your recognition of the improvements made to the manuscript and your thoughtful suggestion regarding the final paragraphs of the Discussion section. In response to your comment, we have carefully reviewed the relevant sections and added several recent and relevant references to support our theoretical interpretations and strengthen the foundation of our claims.
Specifically, we have incorporated the following citations into the last three paragraphs of the Discussion:
-
Emshoff et al. (2024) [Ref. 23]
-
Nguyen-Feng et al. (2020) [Ref. 24]
-
Al-Hussain (2024) [Ref. 25]
-
Enin et al. (2022) [Ref. 27]
-
Trajano et al. (2020) [Ref. 26]
These sources offer empirical support for the psychological impact of voice disorders, the association with social anxiety, and the role of psychosocial interventions in patient care.
We believe that the inclusion of these additional references has considerably enhanced the academic robustness and theoretical grounding of the Discussion. The changes have been clearly marked in the revised manuscript.
Thank you once again for your valuable input, which has helped us improve the quality and clarity of our work.
Reviewer 2 Report
Comments and Suggestions for Authors
I have carefully reviewed the authors’ revisions and their responses to the reviewers’ comments.
They have addressed all the issues raised in a clear and thorough manner, and I commend them for the quality of the improvements made.
In my opinion, the manuscript is now suitable for publication in Behavioral Sciences.
Comments on the Quality of English LanguageI have carefully reviewed the authors’ revisions and their responses to the reviewers’ comments.
They have addressed all the issues raised in a clear and thorough manner, and I commend them for the quality of the improvements made.
In my opinion, the manuscript is now suitable for publication in Behavioral Sciences.
Author Response
Thank you very much for your kind and encouraging remarks.
We greatly appreciate your thoughtful review and are pleased that the revisions and responses met your expectations. Your recognition of our efforts and your recommendation for publication are truly encouraging for us.
Thank you again for your time and valuable contribution to the development of this manuscript.